# HAPPI: Efficient KV cache compression with Hadamard PCA-based Power iteration

## Abstract

Truncated Singular Value Decomposition (SVD) has recently attracted renewed attention for its effectiveness in model optimizations, such as LoRA initialization and KV-cache compression. However, exact SVD remains computationally expensive, while approximate methods like power iteration often introduce non-negligible errors. In this paper, we present Hadamard PCA-based Power Iteration (HaPPI), a new algorithm that significantly improves the accuracy of low-rank approximation while retaining efficiency. Compared to prior methods, HaPPI achieves the lowest approximation error at a practical computational cost. Building on this foundation, we further propose HaPPI-KV, which combines HaPPI with key whitening and residual quantization to deliver high compression ratios for key–value caches. By leveraging both the efficiency and precision of HaPPI, HaPPI-KV achieves state-of-the-art trade-offs between memory efficiency and model quality, highlighting the superiority of our approach.

## 1 Introduction

Truncated Singular Value Decomposition (SVD) has recently re-emerged as a powerful tool for optimizing transformer-based models Wang et al. (2024; 2025a); Yuan et al. (2023); Hsu et al. (2022); Wong et al. (2025); Chen et al. (2021); Sy et al. (2024). By representing data in compact low-rank structures, it significantly reduces storage overhead and has long served as a cornerstone of model compression and lightweight adaptation. Its practical utility is further demonstrated in recent work: initializing LoRA adapters Meng et al. (2024); Yang et al. (2024) with SVD markedly improves fine-tuning accuracy, while exploiting the inherent low rank of key and value matrices enables highly effective KV-cache compression Yankun et al. (2025); Kang et al. (2024); Chang et al. (2025; 2024); Zhang et al. (2024); Wang et al. (2025b); Yan et al. (2025); Lin et al. (2024a). Together, these results underscore the growing importance of truncated SVD in modern optimization.

To advance the quality of low-rank approximation, we propose Hadamard PCA-based Power Iteration (HaPPI), a novel algorithm that achieves precise truncated SVD approximations with practical efficiency. HaPPI builds upon the classical power iteration, a widely used approximate technique, by integrating Hadamard transformations and PCA-inspired initialization. Empirically, HaPPI achieves the lowest per-tensor error among existing methods and recovers fine-tuning performance in LoRA adapters even in scenarios where previous approaches fail due to significant approximation errors.

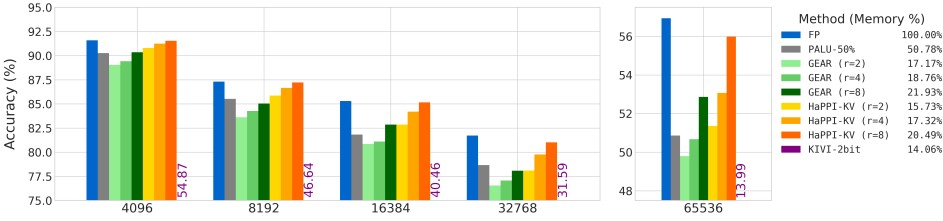

Figure 1: Accuracy across different sequence lengths on Qwen2-7B-Instruct using the RULER dataset. Values in parentheses indicate the relative KV-cache size compared to the FP16 baseline.

Going a step further, we propose HaPPI-KV, a novel method for KV-cache compression. HaPPI-KV extends HaPPI with two additional techniques, key whitening and residual quantization, enabling both rapid compression and state-of-the-art reconstruction quality under tight memory constraints.

In summary, our contributions are threefold: 1) We propose HaPPI, a fast and accurate algorithm for truncated SVD approximation. 2) We provide comprehensive theoretical, quantitative, and empirical validation of HaPPI, including its application to LoRA initialization. 3) We design HaPPI-KV, a high-quality and efficient KV-cache compression technique that outperforms existing approaches. We believe that HaPPI opens a new horizon for model optimization, with HaPPI-KV serving as a compelling demonstration of its potential to advance KV-cache compression.

## 2 RELATED WORK

### 2.1 KV CACHE COMPRESSION

KV cache compression has been widely explored through two main approaches: quantization and low-rank approximation. Quantization-based techniques, such as KIVI Liu et al. (2024b), KVQuant Hooper et al. (2024), and WKVQuant Yue et al. (2024), reduce memory footprint by representing each element with reduced precision. On the other hand, as KV caches are inherently amenable to compression via low-rank structures, several methods leverage this property, including GEAR Kang et al. (2024), PALU Chang et al. (2024), and SVDq Yankun et al. (2025).

Among these, we adopt GEAR as a strong baseline. It combines SVD with quantization, enabling effective KV compression at low cost and without fine-tuning. Our proposed HaPPI-KV enables fast on-demand KV cache compression, while achieving outstanding compression ratios.

### 2.2 TRUNCATED SVD-BASED METHODS

Beyond KV cache compression, truncated SVD plays a crucial role in various optimization tasks, such as weight compression for inference (e.g., SVD-LLM Wang et al. (2024), SVD-LLMv2 Wang et al. (2025a)) and LoRA initialization (e.g., PiSSA Meng et al. (2024), CorDA Yang et al. (2024)). The accuracy of the low-rank approximation significantly influences the final performance of these techniques. For instance, PiSSA reports that LoRA adapters initialized with a naive approximation algorithm exhibit degraded quality after fine-tuning compared to those initialized using an exact truncated SVD. The fast and precise approximation provided by HaPPI enhances the practicality of such methods as well, as we will demonstrate in Section 5.3.

### 2.3 SVD APPROXIMATION METHODS

To make low-rank decomposition computationally affordable, many approximation techniques have been proposed, typically based on randomized initialization Halko et al. (2011), power iteration Vogels et al. (2019), or a combination of both. In this work, we present an improved power iteration algorithm that introduces a novel initialization strategy leveraging the Hadamard transform and PCA. This approach significantly reduces approximation error compared to existing methods, thereby advancing the accuracy–efficiency trade-off in SVD approximation.

## 3 PRELIMINARY

### 3.1 SINGULAR VECTOR DECOMPOSITION

SVD is a fundamental tool for low-rank approximation, where any matrix $A \in \mathbb{R}^{m \times n}$ can be decomposed as $A = USV^T$ with $U \in \mathbb{R}^{m \times m}$ and $V \in \mathbb{R}^{n \times n}$ orthonormal, and $S = \mathrm{diag}(s) \in \mathbb{R}^{m \times n}$ containing singular values in descending order. If the top $r$ singular values dominate, $A$ can be approximated by $A_r = U_{[:,1:r]} S_{1:r,1:r} V_{[:,1:r]}^T$, with the error given by $\|A - A_r\|_F^2 = \sum_{i=r+1}^{\min(m,n)} s_i^2$.

In many recent studies, the idea of truncated SVD has been widely adopted, requiring only the top $r$ singular components. However, applying the exact SVD incurs a computational cost of $O(mn^2)$, as the entire decomposition must be computed before extracting the relevant components.

**Algorithm 1** Power Iteration (Kang et al. (2024))

**Require:** Matrix $\mathbf{A} \in \mathbb{R}^{m \times n}$, Rank $r$, Loop $L$
**Ensure:** Vector $\mathbf{P} \in \mathbb{R}^{n \times r}$, $\mathbf{Q} \in \mathbb{R}^{m \times r}$
1: Initialize $\mathbf{P}$ with random distribution.
2: Initialize $\mathbf{Q}$ with random distribution.
3: **for** $l = 0$ to $L - 1$ **do**
4:    **if** $l = L - 1$ **then**
5:       $[\mathbf{P}, \sim] \leftarrow \text{QR}(\mathbf{P})$
6:    **end if**
7:    $\mathbf{Q} \leftarrow \mathbf{AP}$
8:    **if** $l = L - 1$ **then**
9:       $[\mathbf{Q}, \sim] \leftarrow \text{QR}(\mathbf{Q})$
10:   **end if**
11:   $\mathbf{P} \leftarrow \mathbf{A}^T \mathbf{Q}$
12: **end for**
13: **return** $\mathbf{P}, \mathbf{Q}$

**Algorithm 2** HaPPI (Ours)

**Require:** Matrix $\mathbf{A} \in \mathbb{R}^{m \times n}$, Rank $r$, Loop $L$
**Ensure:** Vector $\mathbf{P} \in \mathbb{R}^{n \times r}$, $\mathbf{Q} \in \mathbb{R}^{m \times r}$
1: $\mathbf{A_H} \leftarrow \mathbf{AH}$
2: $\mathbf{C_H} \leftarrow \mathbf{A_H}^T \mathbf{A_H}$
3: $\mathbf{P}, \_ \leftarrow PowerIter(\mathbf{C_H}, r, min(2, L))$
4: **for** $l = 0$ to $L - 1$ **do**
5:    **if** $l = L - 1$ **then**
6:       $[\mathbf{P}, \sim] \leftarrow \text{QR}(\mathbf{P})$
7:    **end if**
8:    $\mathbf{Q} \leftarrow \mathbf{A_H P}$
9:    **if** $l = L - 1$ **then**
10:      $[\mathbf{Q}, \sim] \leftarrow \text{QR}(\mathbf{Q})$
11:   **end if**
12:   $\mathbf{P} \leftarrow \mathbf{A_H}^T \mathbf{Q}$
13: **end for**
14: **return** $\mathbf{H}^{-1}\mathbf{P}, \mathbf{Q}$

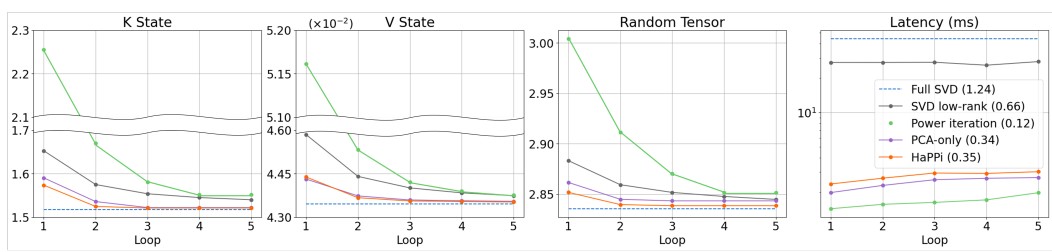

Figure 2: MSE per loop counts of various low-rank approximation methods. K, V state (left, center) is 10th layer of Llama3-8B-Instruct. Note that PCA-only is HaPPI without Hadamard Transformation, and a number next of each method represents GFLOPs.

## 3.2 Hadamard Transformation

Many recent studies Xi et al. (2023); Lin et al. (2024b); Tseng et al. (2024); Kim et al. (2025); Ashkboos et al. (2024); Liu et al. (2024a); Ashkboos et al. (2025); Xiang & Zhang (2024) empirically demonstrated that HT is highly beneficial to mitigate the influence of activation outliers by transforming the data into frequency domain. From a similar perspective, our method also adopt the Hadamard Transform ( Sylvester (1867), HT) as a core building block of HaPPI.

HT is a discrete Fourier-like transform that linearly projects a vector into the frequency domain. Given an $n$-dimensional vector $v \in \mathbb{R}^{2^d}$, its frequency-domain representation $\tilde{v} \in \mathbb{R}^{2^d}$ is obtained by multiplying it with the Walsh–Hadamard matrix $H_d$ as $\tilde{v} = H_d \cdot v$. Here, the Walsh–Hadamard matrix $H_d$ is defined recursively as:

$$H_1 = \frac{1}{\sqrt{2}} \begin{bmatrix} 1 & 1 \\ 1 & -1 \end{bmatrix}, \quad H_n = H_1 \otimes H_{n-1}, \tag{1}$$

where $\otimes$ denotes the Kronecker product, and $H_d$ is a $2^d \times 2^d$ orthogonal matrix satisfying $H_d H_d^T = H_d^T H_d = I$. This orthogonality guarantees the lossless reconstruction of the transformed vector.

## 4 Hadamard PCA based Power Iteration

Efficient low-rank approximation is a critical prerequisite for practical model compression; however, despite extensive research efforts, existing approaches fail to achieve sufficient accuracy under tight computational budgets. A widely used baseline is PyTorch's svd_lowrank, which employs randomized SVD Halko et al. (2011) to generate an initial estimate and optionally refines the result via block power iteration. While this approach achieves relatively low approximation error, it suffers from considerable computational cost and elevated latency. Another notable approach is the power-iteration-based method introduced in PowerSGD Vogels et al. (2019), later adopted by GEAR for

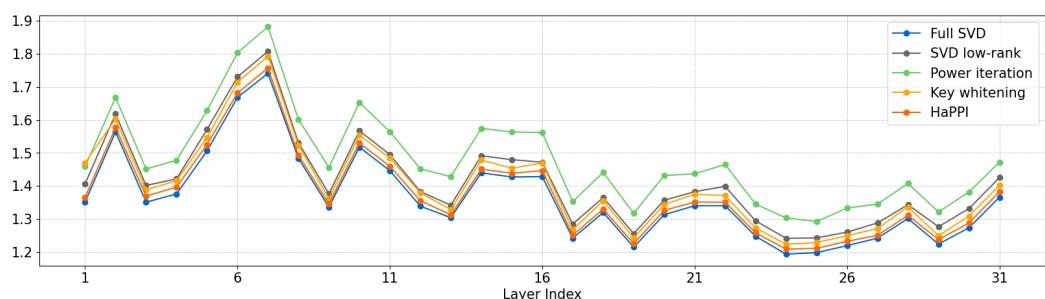

Figure 3: MSE comparison per layers of Llama3-8B-Instruct with GSM8K datasets.

efficient KV-cache compression. As summarized in alg. 1, this method is designed to rapidly decompose the input matrix, but it also exhibits significant approximation errors. Although these methods approximate SVD at substantially lower cost compared to exact decomposition, they exhibit significant residual error even after numerous iterations, as shown in Fig. 2. Such errors often propagate to downstream tasks, resulting in non-negligible quality degradation.

To address this gap, we propose Hadamard PCA-based Power Iteration (HaPPI), a novel algorithm specifically designed to deliver high-quality low-rank approximations with minimal overhead, thereby substantially improving the effectiveness of truncated SVD–based compression pipelines. Alg. 2 presents our proposed HaPPI. While following the general iterative structure of alg. 1, HaPPI introduces two key modifications: (i) an improved initialization procedure (lines 1–2) and (ii) an enhanced final reconstruction stage (line 14). The central ideas are as follows. First, we apply a Hadamard transform to the input tensor prior to decomposition. This transformation projects the tensor into the frequency domain, thereby mitigating the influence of outliers and improving robustness to compression-induced errors, an essential property for transformer tensor compression.

Second, rather than initializing with random vectors, HaPPI computes the covariance matrix of the Hadamard-transformed tensor $A_H$ and uses its decomposed singular vectors as initialization. These left singular vectors can be interpreted as a linear transformation that aligns the original data with the top-$r$ principal axes. While the resulting vectors do not exactly match those of $A_H$ due to the approximate nature of power iteration, they provide an effective starting point that accelerates convergence. After iterative refinement, the resulting singular vectors are mapped back to the original space by multiplying with the inverse Hadamard transform.

## 5 ANALYSIS FOR HaPPI

To demonstrate the superiority of HaPPI, we provide diverse experimental results.

### 5.1 MSE COMPARISON

First, we conducted a series of Mean Squared Error (MSE) comparisons across different low-rank approximation methods (Fig. 2). The evaluation used real K and V tensors extracted from the 11th attention head of the Llama3-8B-Instruct model Grattafiori et al. (2024) on the GSM8K dataset Cobbe et al. (2021), along with a random tensor of identical shape ($802 \times 128$) to assess generalization under a rank-2 approximation. Additionally, Fig. 3 presents layer-wise MSE results across all 32 layers of the model. We compared five approaches: (1) Full SVD (torch.svd), representing the theoretically most accurate baseline; (2) SVD low-rank (torch.svd_lowrank), a memory-efficient approximation variant; (3) power iteration proposed in GEAR (4) PCA-only, which uses principal vectors of the covariance matrix without HT; and (5) HaPPI, the proposed scheme.

As illustrated in Fig. 2, HaPPI achieves approximation results that are consistently closer to the ground-truth SVD solution compared to prior methods. While GEAR's power iteration exhibits the lowest per-iteration computational overhead, its error remains stagnant even with additional iterations. In contrast, HaPPI not only converges more rapidly to a lower error level than svd_lowrank but also achieves substantially improved final accuracy in tensor approximation.

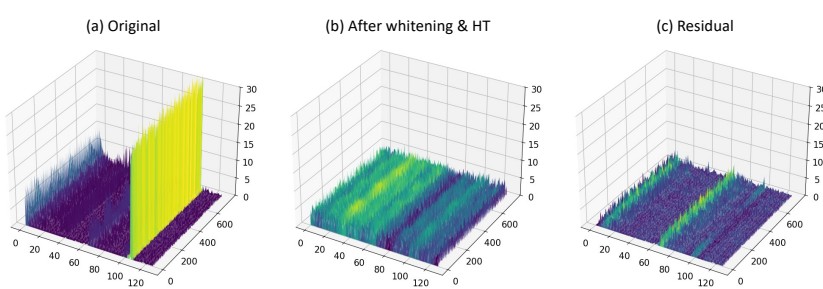

Figure 4: 3D plot of K state of layers.0.attention of Llama3-8B-Instruct.

For the key tensor, having large outliers, HaPPI achieves significantly lower MSE, clearly demonstrating the effectiveness of Hadamard-enhanced PCA initialization. For the random tensor, HaPPI achieves MSE nearly identical to that of full SVD, indicating that its high approximation accuracy generalizes beyond specific data distributions. Most notably, the layer-wise analysis confirms that HaPPI achieves the lowest MSE across all 32 layers, with its performance curve almost overlapping those of Full SVD and SVD low-rank as HaPPI delivers outputs remarkably close to the optimum.

## 5.2 OVERHEAD ANALYSIS

In this section, we analyze the computational overhead of HaPPI in terms of FLOPs, with the results summarized in Fig. 2. Experiments were conducted using a tensor of size (1, 8, 2048, 128), simulating the scenario of processing a sequence of 2048 tokens in Llama3-8B. The number of iterations for all methods except SVD was fixed at two. As shown in the figure, HaPPI requires 0.35 GFLOPs, which is slightly higher than GEAR's power iteration (0.12 GFLOPs) but significantly lower than SVD low-rank (0.66 GFLOPs) and full SVD (1.24 GFLOPs).

This additional cost primarily arises from two sources: HT and the covariance matrix computation. HT introduces minimal overhead using the Fast Walsh–Hadamard Transform, while the primary computational cost comes from computing the covariance matrix $C_H = A_H^T A_H$, which accounts for most of the additional 0.23 GFLOPs. However, as shown by the latency measurements in Fig. 6, despite the FLOPs difference, HaPPI's overall compression time is less expensive than torch.svd_lowrank, close to the latency of power iteration.

## 5.3 VALIDATION ON LoRA FINETUNING

Recent studies have reported that initializing LoRA adapters with truncated SVD leads to improved accuracy. A representative example is PiSSA Meng et al. (2024), which further demonstrates that a noticeable quality degradation arises between exact SVD and its approximation using torch.svd_lowrank. Building on this, we applied HaPPI within the PiSSA pipeline and observed that most of the accuracy degradation caused by approximate SVD was effectively recovered (Tab. 1). This result indicates that the proposed HaPPI algorithm not only reduces numerical error but also provides accurate low-rank approximations that translate into tangible improvements.

| Method | Accuracy |
|---|---|
| LoRA | 44.19 |
| PiSSA | 45.89 |
| PiSSA (SVD low-rank) | 45.08 |
| HaPPI | **45.43** |

Table 1: Average accuracy of LLM finetuning tasks. Refer to appendix for the full table.

## 6 HaPPI-KV

Building on the proposed HaPPI algorithm, we introduce HaPPI-KV, a KV-cache compression technique that realizes the motivation behind prior work and achieves high compression ratio. HaPPI-KV is composed of two key components: key whitening and residual quantization.

## 6.1 KEY WHITENING

In the KV-cache compression process, the key tensor exhibits a distinctive distribution characterized by extreme outliers (Fig. 4 (a)). These outliers degrade the quality of low-rank approximation, making it essential to mitigate their imbalance before compression.

To address this issue and improve the compressibility of the key tensor, we propose a key whitening strategy designed to suppress the adverse effects of outliers. Whitening is a preprocessing step that transforms the covariance structure of data into an identity matrix, normalizing all dimensions to have equal variance. Previous studies Wang et al. (2024; 2025a) have applied whitening for input activation during weight compression. Our approach extends this concept to the key tensor and leverages the efficiency of HaPPI to realize truncated SVD at practical cost, enabling a new lightweight compression method that minimizes outlier-induced errors.

During KV-cache compression, both the key and value tensors are perturbed. To minimize the error introduced in the attention mechanism, it is crucial that the compressed key produces outputs as close as possible to those produced by the original key when computing attention scores for a given query. This objective can be formulated as minimizing the difference between $QK^T$ and $QK'^T$, where $Q, K \in \mathbb{R}^{l \times d}$ represent the query and key tensor, $l, d$ mean sequence length and hidden dimension, respectively, and $K'$ for the compressed Key.

Instead of directly compressing the key tensor with HaPPI, we introduce a whitening matrix $S$ to alleviate channel-wise outliers. With whitening, the target optimization objective becomes:

$$\min \left( \left\| QK^T - QS\,HaPPI(S^{-1}K'^T) \right\|_F \right) \tag{2}$$

Achieving ideal whitening requires the transformed key tensor to have unit covariance, i.e., $(S^{-1}K^T)(S^{-1}K^T)^T = I$, which leads to the condition $K^TK = SS^T$. Applying SVD to $K^TK$ yields $K^TK = US^2U^T$, where $U$ and $S$ are the left singular vectors and singular values, respectively. This implies that $S = US$, indicating that normalization can be achieved by projecting the key tensor along its principal axes. This whitening step effectively suppresses the influence of outliers while preserving the essential structural information of the key tensor.

However, performing full SVD is impractical for compressing KV caches, which are generated dynamically during inference. To address this, we propose a new algorithm that retains the same theoretical foundation but operates at practical computational cost (alg. 3). The key idea is to approximate the principal eigenvectors of the covariance matrix $K^TK$ instead of computing a full SVD. We use a rank $r_1$ larger than the final compression rank $r_2$ used in HaPPI (empirically, $r_1 = 4r_2$ yields the best performance).

---

**Algorithm 3** HaPPI with Key whitening

**Require:** Key $\mathbf{K} \in \mathbb{R}^{l \times d}$, Rank $r_1, r_2$, Loop $L$
**Ensure:** Compressed Key $\mathbf{K}' \in \mathbb{R}^{l \times d}$
1: $\mathbf{C} \leftarrow \mathbf{K}^T\mathbf{K}$
2: $\mathbf{P_{C, \_}} \leftarrow HaPPI(\mathbf{C}, r_1, min(2, L))$
3: $\mathbf{K_C} \leftarrow \mathbf{KP_C}$
4: $\mathbf{P}, \mathbf{Q} \leftarrow HaPPI(\mathbf{K_C}, r_2, L)$
5: $\mathbf{K'_C} \leftarrow \mathbf{QP^T}$
6: $\mathbf{K}' \leftarrow \mathbf{K'_C}\mathbf{P_C^T}$
7: **return** $\mathbf{K}'$

---

The extracted principal components $P_C$ are then used to transform the original matrix $K$. HaPPI is then applied in this transformed space to perform low-rank approximation. In the reconstruction phase, the compressed result is mapped back to the original space by multiplying with $P_C^T$. This process mitigates outlier-induced degradation while preserving the original data scale and structure. From an implementation perspective, both PCA whitening and HaPPI require covariance matrix computation, which can lead to redundant operations. To avoid this inefficiency, we first compute the covariance matrix $C$ and then apply the Hadamard transform to obtain $C_H$. This approach reduces computational overhead by reusing $C$.

Experimental results demonstrate that the proposed PCA whitening strategy effectively addresses the uneven variance of the key tensor and significantly enhances HaPPI's compression performance. As shown in Fig. 4 (b) and (c), the residual tensor after whitening exhibits a far more uniform and compression-friendly distribution, which directly translates to improved final compression quality.

## 6.2 OVERALL COMPRESSION PIPELINE

Fig. 5 illustrates the overall compression pipeline of HaPPI-KV. The core idea is to integrate low-rank approximation with residual quantization for maximum compression efficiency. First, key whitening is applied to the key tensor to mitigate the impact of outliers, while the value tensor exhibits a relatively uniform distribution and therefore does not require whitening.

Next, HaPPI-based low-rank approximation is performed on both key and value tensors. The decomposed results are stored, denoted as $A_k$, $B_k$ for the key and $A_v$, $B_v$ for the value. Subsequently, KIVI

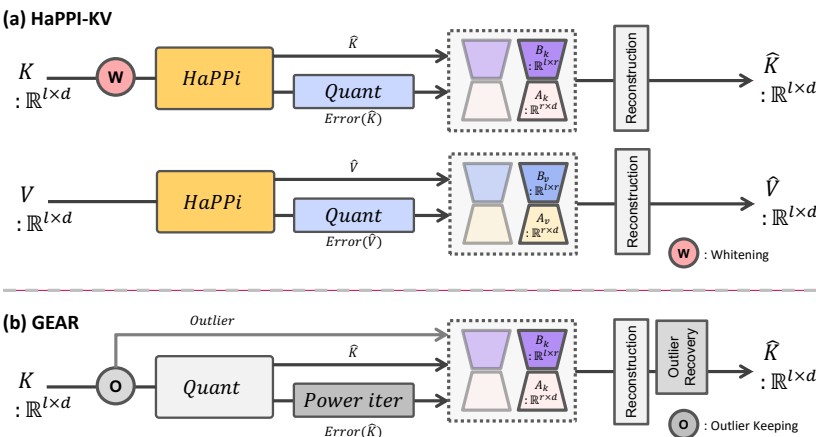

Figure 5: Overview of the pipeline of HaPPI-KV and GEAR.

quantization is applied to the residual tensors. Because the combination of key whitening and HaPPI effectively suppresses outliers, 2-bit group-wise per-token or per-channel quantization is sufficient.

Compared to the most relevant method, GEAR, HaPPI-KV introduces two key innovations. First, it does not rely on excluding outliers prior to compression; the randomized data access complicates implementation and introduces slowdown. Instead, it resolves them through whitening and HT. Second, it applies quantization to the residuals after low-rank approximation. This design makes the residual tensor quantization-friendly, enabling high expressivity with 2-bit data.

# 7 ANALYSIS FOR HaPPI-KV

In this section, we conduct extensive experiments across various language models and benchmarks to comprehensively evaluate the effectiveness of the HaPPI-based KV-cache compression.

## 7.1 EXPERIMENTAL SETUP

We select three widely used open-source models, Llama3-8B-Instruct Grattafiori et al. (2024), Mistral-7B Jiang et al. (2024), and Qwen2-7B Team (2024). For these three models, we evaluate diverse tasks for three major categories. For mathematical reasoning, we use GSM8K Cobbe et al. (2021) and AQuA Ling et al. (2017); for symbolic reasoning, we adopt BigBench Hard (BBH) Suzgun et al. (2022). All reasoning experiments employ Chain-of-Thought (CoT) prompting Wei et al. (2022), which generates extended reasoning traces and provides a rigorous setting to evaluate compression under complex inference workloads.

For long-context understanding, we use LongBench Bai et al. (2023), which includes tasks such as document understanding, summarization, and question answering, allowing us to test HaPPI's performance in realistic application scenarios. Additionally, we leverage RULER Hsieh et al. (2024) to systematically analyze accuracy, memory usage, and latency across a range of sequence lengths, verifying that HaPPI consistently delivers effective compression from short to very long contexts.

All experiments are conducted on NVIDIA A100-SXM4 GPUs (40GB and 80GB). Details of the experimental setup and hyperparameters are provided in the supplementary material.

## 7.2 RESULT ON LLM INFERENCE

To comprehensively evaluate the proposed HaPPI-KV methodology, we conducted a series of reasoning experiments on three representative language models, assessing both Chain-of-Thought (CoT) reasoning and long-context understanding capabilities. The results are summarized in Tab. 2.

In CoT reasoning tasks, HaPPI-KV achieved state-of-the-art performance on most benchmarks. For example, on the Llama3-8B-Instruct model, HaPPI-KV (r=4) reached 78.08% accuracy on GSM8K,

Table 2: Experimental results of LLM inference with various long context datasets. Note that GEAR and HaPPI is quantized to 2-bit integer.

| (%) | | Chain of Thought | | | LongBench | | | | | |
|---|---|---|---|---|---|---|---|---|---|---|
| Model | Method | GSM8K | AQuA | BBH | Single | Multi | Sum. | Few-shot | Synth. | Code |
| Llama3-8B -Instruct | FP | 78.01 | 50.79 | 58.62 | 29.93 | 32.14 | 23.41 | 57.78 | 47.12 | 54.09 |
| | KCVT-3bit | 58.23 | 35.43 | 43.70 | 16.50 | 15.19 | 20.56 | 48.47 | 4.77 | 40.70 |
| | KIVI-2bit | 58.38 | 30.31 | 38.10 | 21.22 | 19.22 | 20.66 | 52.97 | 7.94 | 40.86 |
| | PALU-50% | 71.87 | 43.70 | 54.32 | 30.79 | 32.68 | 22.80 | 57.76 | 45.99 | 53.79 |
| | GEAR (r=4) | 75.89 | 47.64 | 52.35 | 29.24 | 32.51 | 23.53 | 57.85 | 47.06 | 53.97 |
| | GEAR (r=2) | 73.69 | 45.47 | 52.04 | 29.20 | 32.61 | 23.44 | **57.87** | 47.50 | 53.91 |
| | HaPPI-KV (r=4) | **78.08** | **50.00** | **55.27** | 30.99 | 32.38 | **23.82** | 57.28 | 47.06 | **52.73** |
| | HaPPI-KV (r=2) | 77.56 | 48.82 | 54.11 | **31.44** | **32.44** | 23.24 | 57.02 | **47.67** | 52.42 |
| Mistral-7B -Instruct | FP | 55.95 | 33.46 | 47.74 | 39.02 | 37.40 | 26.20 | 63.05 | 66.67 | 59.03 |
| | KCVT-3bit | 44.81 | 33.86 | 42.84 | 32.12 | 25.79 | 25.37 | 58.88 | 43.94 | 52.36 |
| | KIVI-2bit | 41.09 | 25.20 | 35.88 | 31.74 | 29.16 | 24.77 | 57.14 | 28.48 | 52.86 |
| | PALU-50% | 54.13 | **37.80** | 46.56 | 39.30 | 37.56 | 25.38 | 62.97 | 66.67 | 58.87 |
| | GEAR (r=4) | 54.06 | 22.05 | 42.80 | 38.92 | 37.33 | 25.84 | 62.97 | 66.50 | 58.92 |
| | GEAR (r=2) | 53.15 | 24.80 | 42.04 | 38.74 | 37.12 | 25.75 | **63.02** | 66.50 | 59.07 |
| | HaPPI-KV (r=4) | **56.16** | 33.83 | **47.25** | **38.92** | **37.97** | **26.18** | 62.52 | **66.67** | **59.94** |
| | HaPPI-KV (r=2) | 55.63 | 32.29 | 45.10 | 38.90 | 37.87 | 25.89 | 62.35 | 63.31 | 59.62 |
| Qwen2-7B -Instruct | FP | 71.65 | 57.09 | 50.63 | 38.42 | 15.37 | 25.56 | 63.76 | 51.42 | 59.42 |
| | KCVT-3bit | 65.05 | 56.69 | 40.47 | 31.95 | 16.66 | 23.40 | 57.75 | 18.81 | 50.80 |
| | KIVI-2bit | 57.77 | 45.67 | 37.61 | 31.52 | **19.74** | 22.45 | 55.54 | 15.55 | 47.77 |
| | PALU-50% | 65.43 | 57.48 | 48.31 | 38.05 | 15.72 | 24.98 | 63.57 | 51.00 | 59.17 |
| | GEAR (r=4) | 66.11 | 53.94 | 46.29 | 38.51 | 15.15 | 24.78 | 63.69 | 50.92 | **59.36** |
| | GEAR (r=2) | 68.76 | 53.94 | 45.54 | 38.35 | 15.11 | 24.73 | 63.66 | 51.00 | 59.11 |
| | HaPPI-KV (r=4) | **72.93** | **56.70** | **50.07** | 38.59 | 15.34 | **25.61** | 63.74 | 51.33 | 59.21 |
| | HaPPI-KV (r=2) | 71.10 | 55.18 | 49.74 | **38.59** | 15.11 | 25.49 | **63.79** | **51.67** | 59.11 |

nearly matching the original full-precision performance (78.01%) while other compression methods suffered significant degradation. On AQuA and BigBench Hard, HaPPI-KV (r=4) achieved 50.00% and 55.27%, outperforming GEAR (47.64% and 52.35%, respectively).

The advantages of HaPPI-KV became even more evident in LongBench evaluations. On the Llama3 model, HaPPI-KV (r=2) achieved 31.44% accuracy in Single Document QA while reducing memory usage by 84.96%, surpassing the original performance (29.93%). It also maintained 32.44% accuracy in Multi Document QA, nearly identical to the baseline. This suggests that HaPPI-KV's compression process may introduce a regularization effect, slightly improving generalization in certain tasks.

HaPPI-KV consistently demonstrated strong performance across synthetic and code-related tasks as well, almost perfectly preserving original model accuracy. These results confirm that HaPPI-KV offers balanced and robust performance across diverse reasoning domains.

## 7.3 RESULT ON VARIOUS SEQUENCE LENGTH

To systematically evaluate the performance of HaPPI-KV across different sequence lengths, we conducted experiments on the RULER dataset with sequence lengths ranging from 4k to 64k tokens.

The results are presented in Fig. 6. HaPPI-KV consistently outperforms existing compression methods across all sequence lengths. At 4k tokens, HaPPI-KV (r=4) achieved 91.24% accuracy with only a 0.35% drop from the original model, significantly surpassing GEAR (89.42%) and PALU-50% (90.26%). HaPPI-KV (r=2) also achieved 90.8%, outperforming GEAR (r=2) (89.04%).

This advantage becomes more pronounced with longer contexts. At 32k tokens, HaPPI-KV (r=4) achieved 79.76%, outperforming GEAR (r=4) (77.10%) and PALU-50% (78.51%). Even at 64k tokens, it maintained 53.07%, consistently exceeding all competing methods. Overall, HaPPI delivers a 2–3% accuracy improvement over GEAR under identical rank settings.

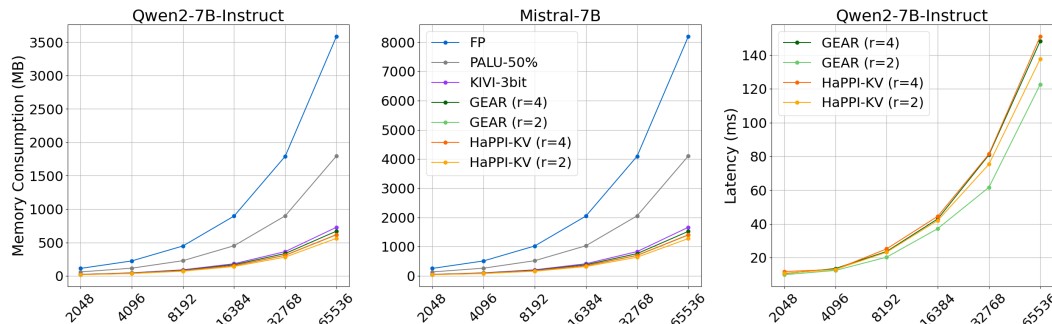

Figure 6: The memory consumption (left and middle) on Qwen2-7B and Mistral 7B, Latency of the KV cache compression function (right) on Qwen2-7B per each sequence length.

## 7.4 MEMORY COMPARISON

Memory usage is a critical factor in evaluating the practicality of KV-cache compression. We measured memory consumption across various sequence lengths for Qwen2-7B and Mistral-7B, as shown in Fig. 6 (left and middle).

HaPPI-KV (r=2) demonstrated the highest memory efficiency across all configurations. At 2,048 tokens on Qwen2-7B, HaPPI used only 19.35 MB, achieving 82.7% memory savings compared to the original, and outperforming GEAR (r=4) and KCVT-3bit. At 65,536 tokens, HaPPI-KV (r=2) used 614.2 MB, saving 82.87% compared to the original's 3,584 MB, significantly outperforming GEAR (r=4)'s 670.31 MB and achieving nearly 3× the efficiency of PALU-50%'s 1,795.5 MB. The higher memory usage of PALU stems from its selective compression strategy, which retains 50% of recent tokens in their original form.

All methods exhibit linear memory growth proportional to sequence length — an inherent characteristic of KV-cache storage. However, HaPPI consistently shows a shallower growth slope, highlighting its scalability advantages in long-context scenarios.

## 7.5 LATENCY COMPARISON

To evaluate computational efficiency alongside memory savings, we measured compression latency for the Qwen2-7B-Instruct attention mechanism, as shown in Fig. 6 (right).

HaPPI-KV (r=4) exhibited latency comparable to existing power-iteration-based approaches. At 2,048 tokens, HaPPI-KV recorded 11.83 ms, nearly identical to GEAR (10.83 ms). At 32,768 and 65,536 tokens, HaPPI-KV recorded 81.43 ms and 148.26 ms, respectively, only less 1% slower than GEAR (80.96 ms and 81.43 ms).

Overall, HaPPI-KV achieves near-equivalent computational efficiency to power-iteration-based methods while delivering significant improvements in both accuracy and memory efficiency. The additional overhead from the Hadamard transform and covariance matrix computation constitutes only a minor fraction of the total cost, demonstrating HaPPI-KV as a practical and scalable solution.

## 8 CONCLUSION

In this work, we proposed HaPPI, a novel Hadamard PCA-based power iteration algorithm that significantly improves the accuracy of truncated SVD approximation while retaining computational efficiency. Building upon this foundation, we further introduced HaPPI-KV, an advanced KV-cache compression framework that combines key whitening and residual quantization to achieve state-of-the-art trade-offs between memory savings, model quality, and latency. Extensive experiments on multiple large language models and diverse benchmarks demonstrated that our method consistently outperforms existing approaches across a wide range of settings. We believe that HaPPI and HaPPI-KV pave the way for future research on scalable and efficient model compression, facilitating the practical deployment of large language models under stringent resource constraints.

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

APPENDIX

## A    EXPERIMENT HYPERPARAMETERS

Table 3: Hyperparameters of Methods.

| Parameters | KCVT-nbit | KIVI-nbit | PALU-50% | GEAR-r=m,Q=nbit | HaPPi-r=m,Q=nbit |
|---|---|---|---|---|---|
| Quantize Bit | n | n | N/A | n | n |
| Group Size | 64 | 64 | N/A | 64 | 64 |
| Key Rank | N/A | N/A | 96 | m | m |
| Value Rank | N/A | N/A | 96 | m | m |
| Loop | N/A | N/A | 3 | 3 | 3 |
| Left | N/A | N/A | N/A | 0.01 | N/A |
| Streaming | True | True | True | True | True |
| Streaming Gap | 64 | 64 | 64 | 64 | 64 |

Table 4: Description of Hyperparameters

| Parameter | Description |
|---|---|
| Quantize Bit | Quantize bit. |
| Group Size | Quantize group size. |
| Key Rank | Rank compared with smaller dimension set to K cache. |
| Value Rank | Rank compared with smaller dimension set to V cache. |
| Loop | Loop of SVD solver (Power-iteration). |
| Left | Outlier extraction part compared with total matrix. |
| Streaming | Use streaming mode. |
| Streaming Gap | Iteration length for re-compression. |

Tab. 3 and 4 present the hyperparameters of the methods considered in our experiments. PALU-50% does not employ quantization and therefore does not define either a quantization bit or a group size. Conversely, KCVT-nbit and KIVI-nbit are quantization-only methods without low-rank approximation, and thus do not specify key rank, value rank, or loop parameters. The Left parameter denotes the fraction of the matrix reserved for outlier extraction; for instance, a value of 0.01 indicates that 1% of the matrix is allocated for this purpose. The Streaming parameter governs the compression of newly generated KV caches. Specifically, the cache is retained in full precision for the duration of the Streaming Gap, after which compression is applied using the designated method. If Streaming is disabled, no compression is applied to the newly generated KV caches.

## B    VISION LANGUAGE MODEL CAPTIONING TASK

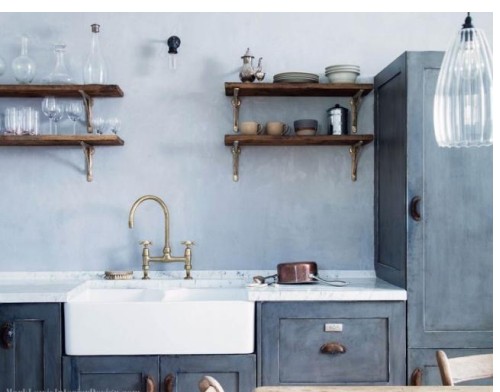

Q: "Please describe this image in detail."

**GEAR-r=2,Q=2bit**

The image features a kitchen with a sink and a dining table. The sink is located on the left side of the kitchen, while the dining table is positioned on the right side. The kitchen is equipped with a shelf above the sink, which is filled with various items such as bottles, bowls, and cups. There are several bottles of different sizes and shapes placed on the shelf, as well as a few bowls and cups. The bottles are arranged in various positions, with some closer to the sink and others further away. The bowls are also placed on the shelf, with one near the sink and **another closer to the dining table**. The cups are scattered across the shelf, **with some near the sink and others near the dining table.**

**HaPPI-KV-r=2,Q=2bit**                                              ✗

The image features a kitchen with a sink and a dining table. The sink is located on the left side of the kitchen, and the dining table is positioned on the right side. The kitchen is equipped with a shelf above the sink, which is filled with various items such as bottles, bowls, and cups. There are several bottles placed on the shelf, with some closer to the sink and others further away. A bowl can be seen on the shelf as well, along with a cup. The dining table is surrounded by chairs, with one chair on the right side and another on the left side. The overall scene gives off a cozy and functional atmosphere.

○

Figure 7: Image captioning results on LLaVA-1.5-7B with GEAR and HaPPI compression methods.

Fig. 7 presents a qualitative evaluation of caption generation on a sample image using the LLaVA-1.5-7B model Liu et al. (2023) with two KV cache compression methods: GEAR (r=2, Q=2bit) and HaPPI-KV (r=2, Q=2bit). At the beginning of the generation process, both GEAR and HaPPI-KV produce nearly identical captions, suggesting that the information is well preserved under compression. However, as the generation proceeds, GEAR tends to induce hallucinations, as illustrated in the figure, whereas HaPPI-KV consistently generates accurate and faithful captions.

## C ABLATION STUDY

Table 5: Ablation study on Mistral-7B with GSM8K-CoT benchmark in r=2, q=2 environment.

| Method | Original | Residual | Additional | Accuracy |
|---|---|---|---|---|
| FP | - | - | - | 55.95 |
| GEAR | Quant. | Power Iteration | Outlier preserving | 53.15 |
| | Quant. | Power Iteration | - | 51.87 |
| | Quant. | HaPPI | - | 53.10 |
| | HaPPI | Quant. | - | 54.13 |
| HaPPI-KV | HaPPI | Quant. | Key whitening | 55.63 |

To systematically analyze the impact of each component of HaPPI-KV on overall performance, we conducted an ablation study using the Mistral-7B model and GSM8K-CoT benchmark.

As shown in tab. 5, the baseline GEAR methodology exhibited a 2.8% performance degradation compared to the FP baseline. Interestingly, when the outlier preserving strategy was removed from GEAR, performance further declined by 1.28% to 51.87%, suggesting that outlier handling plays a crucial role in KV cache compression.

In the experiment where power iteration was replaced with HaPPI, we achieved 53.1% accuracy, confirming a 1.23% performance improvement over the baseline power iteration. This demonstrates that HaPPI's improved low-rank approximation capability is effective in actual downstream tasks. Furthermore, analyzing the impact of component order in the compression pipeline, we found that the strategy of applying HaPPI first followed by quantization achieved 54.13% accuracy, showing superior results compared to the reverse order. This indicates that performing high-quality low-rank approximation first and then applying quantization to the residuals is more effective.

Finally, the complete HaPPI-KV pipeline with key whitening achieved a final performance of 55.63%, obtaining an additional 1.5% performance improvement from key whitening alone. This confirms that the preprocessing step of normalizing the outlier distribution of the key tensor makes a substantial contribution to overall compression quality. Overall, each component cumulatively contributes to performance improvement, ultimately achieving 55.63%, which is nearly equivalent to FP performance, demonstrating the efficacy of the proposed methodology.

## D LORA INITIALIZATION TASK

Table 6: Experimental accuracy (%) results of LoRA finetuning on Llama3-8B-Instruct. Note that rank is 16 and loop count for PiSSA (SVD approx.) is 2.

| Method | GSM8K | MATH | HumanEval | MBPP | IFeval | Average |
|---|---|---|---|---|---|---|
| LoRA | 69.07 | 12.58 | 46.96 | 43.16 | 49.18 | 44.19 |
| PiSSA | 69.67 | **13.20** | 49.75 | **46.45** | 50.36 | **45.89** |
| PiSSA (SVD low-rank) | 69.21 | 12.58 | 49.40 | 45.21 | 49.02 | 45.08 |
| HaPPI | **70.80** | 13.01 | **49.92** | 45.21 | **53.02** | 45.43 |

To validate HaPPI's accurate low-rank approximation capability, we evaluated its performance on LoRA adapter initialization tasks. Recent studies have reported that LoRA initialization using truncated SVD significantly improves fine-tuning performance, and particularly in methodologies like PiSSA, performance differences exist between exact SVD and approximate SVD.

The experimental results shown in tab. 6 represent accuracy across various tasks performed on the Llama3-8B-Instruct model with rank 16 settings. The baseline LoRA methodology recorded an average accuracy of 44.19%, while PiSSA with SVD-based initialization showed a 1.7% performance improvement to 45.89%.

A notable observation is that when PiSSA used approximate SVD instead of exact SVD, performance declined by 0.81% to 45.08%. This demonstrates that SVD approximation errors directly impact actual downstream task performance. In contrast, when HaPPI was applied, we achieved an average accuracy of 46.79%, surpassing all comparison methodologies. Particularly, we recorded 70.8% on GSM8K, 13.01% on MATH, 51.93% on HumanEval, and 53.02% on IFeval, achieving the highest performance on most individual tasks.

These results demonstrate that HaPPI provides high-quality low-rank approximation that leads to actual model performance improvements beyond simply reducing numerical approximation errors. In particular, the 0.9% performance improvement over PiSSA clearly shows the importance of accurate SVD approximation and the practical value of the HaPPI algorithm.

