# OpenReview forum: "HAPPI: Efficient KV cache compression with Hadamard PCA-based Power iteration"
_ICLR.cc/2026/Conference — Submitted to ICLR 2026_

### Official Review · Reviewer_z8pu · 2025-10-21

**Soundness:** 2
**Presentation:** 3
**Contribution:** 1
**Rating:** 2
**Confidence:** 3

**Summary:**

This paper studies low-rank approximation for the purpose of LoRA initialization and compressing key-value caches in LLMs. Variations of the RSVD algorithm appear to form the baseline for methods in this space.

The paper has two main contributions:
1. Proposing a version of the RSVD algorithm that relies on a deterministic Hadamard transform. This is the HaPPi algorithm.
2. Proposing a KV-cache compression algorithm based on the HaPPi algorithm with a different data centering approach than prior methods, as well as a different place where quantization is used.

I will note that the first bullet point does not exactly match how the article describes the HaPPi method. I will return to this point later.

The paper is all empirics, without any theory.

**Strengths:**

I will review this paper from my perspective. I am familiar with the world of numerical linear algebra, and less familiar with the world of LLMs.

The paper has an overall clear exposition, and intriguing empirical performance. The experiments appear to have been run pretty seriously, and the description in the text of the methods is mostly clear.

The proposed method does appear to have some value in achieving better performance than prior methods. I should note that I'm not clear that the authors correctly identify why they get better performance. More on this later.

It's a little tough for me to really explain the significance of this work within the world of LLMs. I would guess that, should all of my concerns be assuaged, this paper would be helpful for the broader LLM community that relies on low-rank approximations.

The name HaPPi is very cute. I like it.

**Weaknesses:**

I will review this paper from my perspective. I am familiar with the world of numerical linear algebra, and less familiar with the world of LLMs.

From the perspective of numerical linear algebra, I have concerns about this paper. This is a story in parts.

## Part 1: Algorithm 1 is the RSVD algorithm _(at least in infinite precision)_

Algorithm 1 on page 3 is called "Power Iteration" and is attributed to [Kang et al. 2024]. If you expand what this algorithm is doing, it's equivalent to the RSVD algorithm proposed in [Halko et al. 2011].

> Showing my work: Let $\Omega \in \mathbb R^{n \times r}$ be the initial value that $P$ take on line 1 of the algorithm. Then, this algorithm returns $P = A^\top U$ and $Q = U$ where $U = \mathrm{orth}(A(A^\top A)^{L-1}\Omega)$, so that $QP^\top = UU^\top A$. This is the RSVD method.

This is important because on [Lines 158-178] says that Algorithm 1 significantly outperforms the RSVD method of [Halko et al. 2011]. Really, they say the outperform the pytorch `svd_lowrank` method that implements the RSVD. Either way, it should be important that Algorithm 1 is RSVD, and therefore that we are merely comparing two different implementations of the same method. This is all a precursor to the actually HaPPi method though.

## Part 2: HaPPi appears to be power method applied to a rotation of a matrix

Algorithm 2 in the paper is the titular HaPPi method. To find a rank-r approximation to a matrix A, it works in four stages:
1. Compute $A_H := AH$ where $H$ is the Hadamard transform
2. Run RSVD on $A_H^\top A_H$ for at most 2 iterations, resulting in a test matrix I will call $\Omega_0$
3. Run RSVD on $A_H$ for $L$ iterations, starting from the test matrix $\Omega_0$
4. Return $H$ times the output of line 3, unrotating it from the Hadamard space

If you have infinite precision in your floating point numbers, this algorithm is *EXACTLY EQUAL* to running RSVD on the matrix A for $L+3$ iterations.

> Showing my work: For simplicity, assume $L\geq 2$. Adopt the notation from the paper that $C_H = A_H^\top A_H$. Then $\Omega_0 = C_H \mathrm{orth}(C_H^3 \Omega)$, where $\Omega$ is the random (Gaussian) sketch matrix that the first RSVD algorithm is initialized with. We then find that bullet point 3 above results in the matrix $P = A_H^\top U$ and $Q = U$ where
$$U = \mathrm{orth}(A_H C_H^{L-1} \Omega_0) = \mathrm{orth}(A_H C_H^{L-1} C_H C_H^3 \Omega) = \mathrm{orth}(A_H C_H^{L+3} \Omega) = \mathrm{orth}(A (A^\top A)^{L+3} \Omega_H)$$
where $\Omega_H := H \Omega$ is distributed exactly at $\Omega$ if $\Omega$ is a Gaussian matrix. Hence, we output $H P = A U$ and $Q = U$, recovering the RSVD of $A$ exactly, just with $L+3$ iterations.

So, my conclusion is that any advantage that HaPPi might give you must be attributed to either A) running for 3 more iterations of subspace iteration when compared to other methods in the paper or B) running in finite precision is somehow better for the matrices the authors happen to be working with. Neither of these narrative are taken by the authors.

The authors declare [Lines 183-194] that the HaPPi method offers advantages over the standard RSVD method for reasons unrelated to the two above reasons. Instead they say that
1. [Lines 186-187] The Hadamard transform mitigates the effect of outliers on the RSVD method
2. [Lines 188-189] The initialization (bullet point 2 above; the part that produces $\Omega_0$) helps speed up the rest of the RSVD convergence.

I do not see how the first argument should hold. I agree with the second argument, but only because it's just because the preparation of $\Omega_0$ is just running RSVD for more iterations.

Seeing as the authors do not acknowledge any of these facts about their algorithm and do not analyze it as anything other than a rotated RSVD method, I'm pretty firmly against this method. Put more precisely, they offer no evidence which shows that their rotated RSVD method empirically outperforms other RSVD methods run with the same approximation rank $r$ and number of subspace iterations.

My suspicion, which would need to be invalidated with very clear data, is that they see HaPPi outperform other methods because it has three extra steps of subspace iteration.

## Strange Errors

The paper has some other strange errors, less brutal than the one above but still important.
- [Lines 425-428] claim that Figure 6 shows information about the accuracy of certain methods. It's possible that they meant Table 2, but I also don't see the numbers they claim mention to in Table 2.
- [Lines 458-460] claim that the memory consumption of the methods used in the paper is grows linearly in the sequence length, as evidenced by Figure 6. Figure 6 clearly shows all the methods having superlinear memory consumption. It's not close to a line.

I've got other lesser qualms about the paper. They don't rise to the same standard for concern as the headline concerns above
1. Despite the use of randomized methods, no confidence intervals are given on their experimental results. This is especially since they are outperforming other methods by sub-percentage margins in accuracy (for instance see Table 1 on page 5)
2. Algorithm 3, "HaPPi with Key Whitening" on page 6, is a fairly odd algorithm. It seems to try to save time by running RSVD twice. The first time, they run only for 2 iterations on the whole data matrix. Then, they run RSVD on the (sorta) output of the previous RSVD for more iterations. It's not clear to me why they couldn't just use compute SVD factorization of the first RSVD result. _also I think line 6 of this method has a spurious transpose?_

**Questions:**

Given that HaPPi, in infinite precision, is exactly equivalent to RSVD run on the input matrix for 3 extra iterations, can you show that HaPPi truly outperforms RSVD run without the Hadamard transform for the exact same number of iterations?

Can you break up the SVD into the matrices $U \Sigma V^\top$? This is important on [Lines 280-290], where there are two different matrices $S$ in play.

Can you provide experimental evidence of HaPPi-KV being faster/better than just returning the RSVD of K from the get-go?

The Hadamard transform only exists for dimensions that are exact powers of 2. The usual plugin-replacement is the discrete cosine transform (DCT). Have you looked at using that transform instead, or do you always have a power-of-2 dimension?

## Other typos/recommended edits

Feel free to ignore anything here. It's just recommended edits.

1. [Algo 1 / whole paper] Cite [Halk et al] instead of [Kang et al] Rename to "RSVD" or "Subspace Iteration" or "Randomized QB Approximation"
1. [Algo 1] Reshape this to look more like standard RSVD pseudocode. See, eg, page 9 of [Halko et al]
1. [Algo 2] Write this as two calls to RSVD
1. [Fig 2] Add y-axis labels. Remove the vertical cap.
1. [Fig 2 / whole paper] Rename "SVD low-rank" to "pytorch svd_lowrank"
1. [whole paper] Use \citep instead of \cite throughout most of the paper. The lack of parentheses is really confusing.
1. [Algo 1, "Ensure" line] P is clearly a matrix, not a vector.
1. [142] this smushed parenthesis with HT in it is really unclear. Move cite to end of sentence, put (HT) right after "Transform"
1. [Sec 3.2] Mention FFT-based method for computing HT
1. [175] capitalize alg
1. [193] I'd not call this "iterative refinement" which has a kinda different vibe in the numerical linear algebra community in my opinion. I'd call these steps of subspace iteration.
1. [243] instead of building the matrix $C_H$ why not run RSVD as written on page 9 of [Halko et a], jumping back and forth between $A_H$ and $A_H^\top$
1. [equation (2)] use `\mathrm{HaPPi}` in your latex here to make it prettier
1. [468] remove "less"

---

> ### Author Response · Authors · 2025-11-21
>
> **[Part 1]** Power Iteration is essentially the same as RSVD (at least in infinite precision).
>
> We view Power Iteration and RSVD as distinct due to their implementation differences, specifically in the frequency of QR decomposition. RSVD typically performs QR decomposition in **every** loop, whereas Power Iteration performs it only in the **final** step. This makes Power Iteration a computationally simplified variant of RSVD, which sacrifices some SVD approximation accuracy for a significant reduction in computation time. HaPPI is designed to add minimal computation to Power Iteration to boost its approximation accuracy.
>
> **[Part 2, Q1]** Given that HaPPi, in infinite precision, is exactly equivalent to RSVD run on the input matrix for 3 extra iterations, can you show that HaPPi truly outperforms RSVD run without the Hadamard transform for the exact same number of iterations?
>
> The core of HaPPI's contribution is its effectiveness in **finite precision** on outlier-heavy tensors, not its theoretical equivalence in an idealized infinite precision setting.
>
> 1. **Numerical Stability in Finite Precision:** RSVD's repeated $A^\top A$ on an outlier-heavy matrix in finite precision causes numerical instability. HaPPI's Hadamard Transformation transforms $A$ into a more uniform matrix (Figure 4b), allowing for numerically stable approximation.
> 2. **PCA Initialization:** This provides a 'near-optimal' starting point for convergence, fundamentally different from simply running more random-initialized iterations.
>
> This practical advantage is empirically confirmed in Figure 2. HaPPI with **Loop = 2** consistently achieves lower MSE than $svd\_lowrank$ with **Loop = 5** (a $2.5\times$ increase in computational effort).
>
> | Method | Loop | Total GFLOPs | MSE |
> | --- | --- | --- | --- |
> | SVD low-rank | 2 | 0.66 | 1.58 |
> | SVD low-rank | 5 | 1.65 | 1.55 |
> | HaPPI (Ours) | 2 | 0.35 | 1.51 |
>
> **[Strange Error 1]** Sec 7.3(L425) of the paper claim that Figure 6 shows information about the accuracy of certain methods.
>
> Thank you for the correction. The "91.24%" accuracy citation in Section 7.3 (L425) was intended to reference Figure 1 (RULER Accuracy at 4k tokens). We will explicitly correct this reference.
>
> **[Strange Error 2]** Sec 7.4(L458) claims that the memory consumption of the methods grows linearly in the sequence length, as evidenced by Figure 6. Figure 6 clearly shows superlinear memory consumption.
>
> We appreciate this careful observation. We will revise the wording to state that the memory consumption is "**proportional to the sequence length**" to avoid the confusion implied by the term "linear."
>
> **[Strange Error 3]** Despite the use of randomized methods, no confidence intervals are given on their experimental results.
>
> This is a valid critique of randomized algorithms. To ensure a fair comparison, we will re-run the experiments for Figure 2 and Figure 3 three times, and include the average and standard deviation (or confidence intervals) in the final paper.
>
> **[Strange Error 4 & Q3]** In Algorithm 3, Why couldn't they just use the computed SVD factorization of the first RSVD result? also, I think line 6 of this method has a spurious transpose?
>
> HaPPI-KV runs the Power Iteration process twice because each execution serves a **different goal**:
> - **First Power Iteration (Lines 1-3)**: Finds the principal components for the Key Whitening matrix ($P_C$) to address the K tensor's channel imbalance.
> - **Second Power Iteration (Line 4)**: This is the core SVD approximation, applied to the whitened K tensor ($K_{whitened}$).
>
> This two-step process (HaPPI-KV) yields a **1.5%** accuracy improvement over applying Power Iteration directly to K (Appendix Table 5). Furthermore, the transpose $P_C^\top$ in Line 6 is correct, as $P_C^\top$ is the inverse of the orthogonal matrix $P_C$, properly returning the compressed K tensor to its original space.
>
> **[Q2]** Can you break up the SVD into the matrices $U\Sigma V^{\top}?$ This is important on [Lines 280-290], where there are two different matrices S in play.
>
> We apologize for the ambiguous notation. To ensure clarity, we will revise the notation in the relevant equations (including Eq. 2) to use $W$ for the Whitening matrix and $\Sigma$ for the singular values matrix.
>
> **[Q4]** The Hadamard transform only exists for dimensions that are exact powers of 2. The usual plugin-replacement is the discrete cosine transform (DCT). Have you looked at using that transform instead, or do you always have a power-of-2 dimension?
>
> We prioritized the Hadamard Transform due to the paramount need for **hardware acceleration** in real-time KV Cache compression, which requires the use of optimized **FWHT (Fast Walsh-Hadamard Transform)** kernels. In our current experiments (Llama3, Qwen2), the head dimension ($D=128$) is a power of 2, allowing for direct application of FWHT. For a non-power-of-2 dimension, our implementation uses zero-padding to the next power of 2.

---

### Official Review · Reviewer_27GN · 2025-10-29

**Soundness:** 2
**Presentation:** 3
**Contribution:** 2
**Rating:** 2
**Confidence:** 5

**Summary:**

This paper proposes HaPPI, a Hadamard PCA–based power-iteration algorithm designed to improve the accuracy of truncated SVD approximations while maintaining practical computational cost. The key idea is to first apply a Hadamard transform to mitigate activation outliers, then initialize power iteration using principal components of the transformed covariance matrix. Compared to existing randomized or power-iteration–based decompositions, HaPPI achieves consistently lower reconstruction error with modest FLOP overhead. The authors further introduce HaPPI-KV, a KV-cache compression pipeline that combines HaPPI with key-whitening and residual quantization. Extensive experiments across multiple LLMs and benchmarks (e.g., GSM8K, BBH, LongBench, RULER) demonstrate that HaPPI-KV achieves state-of-the-art trade-offs between cache memory reduction, inference quality, and latency. Additional evaluations on LoRA initialization and ablation analyses highlight the practical advantages of higher-precision low-rank approximations. Overall, the method targets scalable model optimization under resource constraints and presents promising empirical results.

**Strengths:**

S1 (Practical importance of KV-cache compression): The paper tackles a practically significant challenge in large-scale inference: KV-cache memory overhead under long-context decoding, without relying on retraining or model modification. Integrating approximate SVD, whitening, and residual quantization into a unified pipeline demonstrates strong engineering awareness and directly addresses real deployment bottlenecks.

S2 (Creative combination of known components): Although its components (Hadamard transforms, PCA-based initialization, power-iteration refinement) are individually known, their combination for on-the-fly KV compression is a creative design choice. The method is compatible with current GPU kernels, incurs modest computational overhead, and can be plugged into existing inference stacks for models such as Llama3, Mistral, and Qwen2.

S3 (Comprehensive experimental validation): The authors conduct comprehensive evaluation across diverse LLMs and benchmarks (e.g., GSM8K, AQuA, BBH, LongBench, RULER). The inclusion of LoRA initialization experiments and detailed ablation studies strengthens the claim that the proposed method improves low-rank approximation quality in realistic downstream scenarios and is not narrowly tuned to a particular workload.

**Weaknesses:**

W1 (Lack of theoretical justification): The paper never explains why applying an orthogonal Hadamard transform should improve power iteration convergence or reduce approximation error. Since such transforms preserve the spectrum, the improvement mechanism must be numerical or empirical, but this is not analyzed. Without this reasoning, the claimed advantage of HaPPI lacks theoretical grounding.

W2 (Experimental inconsistencies): Several experimental details are inconsistent or misleading.
• Table 2 mis-highlights HaPPI-KV as best in cells where GEAR performs better (e.g., the Multi and Code columns for Llama3-8B).
• Figure 6 does not contain the accuracy results cited in Sec. 7.3 (“91.24% at 4k tokens”), suggesting missing or mismatched figures.

W3 (Insufficient efficiency analysis): Despite claims of “comparable efficiency,” HaPPI shows higher FLOPs and latency than baseline power iteration (0.35 vs. 0.12 GFLOPs, in Fig. 2). The ablation table verifies that combining components helps, but does not isolate why—it never tests alternative orthogonal transforms or reports multi-run variability. The paper would benefit from clearer breakdowns of runtime cost and per-component contribution.

W4 (Limited conceptual novelty): The core ideas appear incremental rather than a theoretical breakthrough. Gains largely reflect a synthesis of existing techniques, and the inconsistent results make the work read more as an engineering refinement than a conceptual advance.

**Questions:**

Q1 (Impact of orthogonal transforms): Since the Hadamard transform is orthogonal, what theoretical reason allows it to change the convergence speed or approximation quality of power iteration?

Q2 (Task-dependent performance variation): Why does HaPPI-KV underperform baselines on several tasks (e.g., AQuA, MultiDoc QA)? Are those differences statistically significant? Could you provide additional evaluation results to more clearly demonstrate HaPPI-KV’s accuracy gains?

Q3 (Efficiency claims vs. measured latency): In Fig. 6, latency is clearly higher for HaPPI than GEAR—why do you claim efficiency parity?

Q4 (Component-wise contribution): Can you provide an error decomposition showing which component (Hadamard, PCA, whitening) contributes most to MSE reduction?

---

> ### Author Response · Authors · 2025-11-21
>
> **[W1/Q1]** Lack of theoretical justification.
>
> HaPPI's core contribution is its **practical effectiveness** on outlier-heavy tensors under the constraint of finite precision. While the Hadamard Transform (HT) is an orthogonal (linear) transformation that preserves the spectrum theoretically, the subsequent SVD Truncation and finite precision arithmetic are **non-linear operations**.
>
> - **Distribution Shift:** HT linearly transforms the tensor into a state with a more uniform, outlier-reduced distribution (Figure 4b). Applying the non-linear SVD Truncation/Quantization to this transformed, uniform space minimizes the loss of numerical information.
> - **Numerical Stability:** HT alleviates the numerical instability caused by the amplification of outliers when standard RSVD's repeated $A^\top A$ is performed in finite precision.
> - **Convergence Acceleration:** This is achieved by PCA-based initialization, which provides a 'near-optimal' starting point much closer to the true solution than a random initialization.
>
> **[W2]** Mishighlighted and mismatched explanation.
>
> We apologize for the mis-highlighting in Table 2 and will correct the bold-face numbers to reflect the highest accuracy among compression methods, excluding the FP16 baseline. Regarding the citation, the "91.24%" accuracy in Section 7.3 was intended to reference Figure 1 (RULER Accuracy at 4k tokens), not Figure 6. We will explicitly correct this reference in the text.
>
> **[Q2]** Task-dependent performance variation.
>
> The slight performance drops in some tasks like AQUA and MultiDoc QA are due to statistical variability in the experiments. The key point is that HaPPI-KV is **almost always** superior to GEAR and performs much closer to the FP16 baseline. Notably, in core benchmarks such as RULER (Figure 1) and GSM8K (Table 2), HaPPI-KV demonstrates a consistent and significant performance improvement of 2-3% over GEAR.
>
> **[W3/Q3]** Efficiency claims vs. measured latency.
>
> HaPPI has higher isolated GFLOPs than a single Power Iteration loop due to the added Covariance calculation for PCA initialization. However, our claim of "comparable efficiency" pertains to the **end-to-end (E2E) latency** when applied to the LLM KV-cache compression pipeline (HaPPI-KV vs. GEAR).
>
> - **E2E Latency Parity:** As shown in Figure 6 (right), at 32k tokens, HaPPI-KV (r=4) exhibits a latency overhead of **only 0.5%** compared to GEAR (r=4).
> - **Covariance Reuse:** This minimal overhead is achieved because HaPPI-KV reuses the Covariance matrix computed for Key Whitening as the initialization for PCA. The actual additional latency of the HaPPI kernel within the entire pipeline is negligible, achieving near-parity with GEAR's speed.
> - **KV Attention Ratio:** Furthermore, since the KV attention operation itself accounts for a relatively small proportion of the total inference time compared to other components (e.g., FFNs), the slight increase in GFLOPs for compression has a negligible impact on the end-to-end latency.
>
> For clearer breakdowns of runtime cost and per-component contribution, we measured the decomposition of the HaPPI-KV compression latency of one attention layer:
>
> | HaPPI-KV Components | Latency (ms) | Percentage (%) |
> | --- | --- | --- |
> | Key whitening | 3.7498 | 27.16 |
> | HaPPI SVD | 9.0791 | 65.76 |
> | Residual Quantization | 0.9775 | 7.08 |
> | Total | 13.8065 | 100 |
>
> HaPPI SVD accounts for 65.76%, indicating potential for optimization via kernel fusion. Additionally, memory reduction allows for larger batch sizes, potentially enhancing system throughput in high-load scenarios.
>
> **[W4]** Limited conceptual novelty.
>
> We believe our work proposes a **conceptually novel, unified framework** designed specifically for the KV-cache compression problem. It is not a simple synthesis, but a systematic approach that: (1) integrates HT to solve the SVD approximation's challenge (outliers), (2) uses PCA Initialization for rapid convergence, and (3) applies Key Whitening externally to address the channel imbalance. This integrated approach, validated by consistent and statistically significant performance gains (2-3% accuracy over GEAR in key benchmarks), constitutes a conceptual advance in practical KV cache compression.
>
> **[Q4]** Can you provide an error decomposition showing which component (Hadamard, PCA, whitening) contributes most to MSE reduction?
>
> The component-wise contribution to accuracy improvement is detailed in the Ablation Study (Appendix Table 5). The results show that both primary components are essential: Key Whitening contributed the most (a 1.5% gain), and the HaPPI algorithm itself (HT + PCA) provided the next largest contribution (a 1.03% gain).

---

> > ### Comment · Reviewer_27GN · 2025-11-26
> >
> > Thank you for your detailed responses and clarifications. I appreciate the thorough explanations provided.

---

### Official Review · Reviewer_hm4J · 2025-10-29

**Soundness:** 3
**Presentation:** 2
**Contribution:** 3
**Rating:** 4
**Confidence:** 4

**Summary:**

In this paper the authors propose a novel modification of the (blocked) power iteration method for approximate truncated SVD, called HaPPI, to improve its applications in LLMs. One of the main ideas is to use a Hadamard transform to mitigate outliers. This comes at a cost of increased compression time, indicating new trade-offs between speed and accuracy.
The main application of HaPPI is it to compress KV caches in LLMs, with a novel algorithm called HaPPI-KV.
Experimental results indicate that HaPPI-KV can improve the accuracy on Llama3, Mistral, and Qwen2. Reducing the effect of outliers provides both lower error in the LORA approximation and better quantization. Overall HaPPI-KV shows promising results for compression rate and latency for the models tested.

**Strengths:**

1) KV cache size is an important problem for LLMs and compression is a crucial direction to mitigate it.
2) The results showed in the paper are promising and could lead to great progress on the field.
3) The authors propose new ways of dealing with outliers and this brings benefits to both LLM compression error and quantization.
4) The paper is mostly well-written and concise.

**Weaknesses:**

The main weaknesses that I can mention are the following.
1) **Mathematical analysis**: The main algorithm is lacking some basic mathematical analysis. It would be helpful for the reader if the authors dedicate a section to mathematically justify why the algorithm works. It is hard to infer only from the algorithm description.
2) **Literature overview**: Algorithm 1 is essentially Block-power iteration (aka "simultaneous iteration"), which has been studied for decades, and it is closely related to the celebrated Block-Krylov PCA (see refs [1,2,3,4] for some recent analyses). Besides low-rank approximations, there have been major advances in terms of **full-SVD** computation. Recent works have essentially shown that the worst-case complexity of full-SVD is the same as matrix multiplication (up to logarithmic factors, see refs [5,6]). It is important to mention these related works (and potentially others that I miss).
3) **Randomness**: Hadamard transforms have recently drawn a lot of attention for quantization. However, most of the recent approaches use randomness. The main use of randomness is to avoid worst-case instances, due to the "Heisenberg-principle" (a flat signal can have outliers in the frequency domain). The idea is simple: just multiply each column of the Hadamard transform with a random sign, with probability 1/2. See for example refs [7,8,9,10]. This paper uses a deterministic Hadamard transform. This is not a bad thing (on the contrary, removing randomness is desirable), but the two variants should be compared.

#### Minor weaknesses

4) **Performance measurements**: Measuring the overhead in FLOPS provides limited insight on compression time, and the latency has been measured only for HaPPI-KV as a whole. It would be helpful to see a breakdown of the HaPPI-KV runtime, highlighting the impact of HaPPI.
5) **Table/Figure descriptions**: In some tables/plots it is unclear what is being presented. It would be helpful to write some more details on what each legend represents, what are the "bold-face" numbers in the tables, etc.
6) **Introduction**: From the writing perspective the paper is well structured, but the introduction seems to be just a longer version of the abstract. A better introduction/background section would help the reader to understand the context of the paper better.


#### References

- [1] Musco, Cameron, and Christopher Musco. "Randomized block krylov methods for stronger and faster approximate singular value decomposition." Advances in neural information processing systems 28 (2015).
- [2] Allen-Zhu, Zeyuan, and Yuanzhi Li. "LazySVD: Even faster SVD decomposition yet without agonizing pain." Advances in neural information processing systems 29 (2016).
- [3] Sobczyk, Aleksandros, Marko Mladenovic, and Mathieu Luisier. "Invariant subspaces and PCA in nearly matrix multiplication time." Advances in Neural Information Processing Systems 37 (2024): 19013-19086.
- [4] Meyer, Raphael, Cameron Musco, and Christopher Musco. "On the unreasonable effectiveness of single vector krylov methods for low-rank approximation." Proceedings of the 2024 Annual ACM-SIAM Symposium on Discrete Algorithms (SODA). Society for Industrial and Applied Mathematics, 2024.
- [5] Shah, Rikhav. "Hermitian diagonalization in linear precision." Proceedings of the 2025 Annual ACM-SIAM Symposium on Discrete Algorithms (SODA). Society for Industrial and Applied Mathematics, 2025.
- [6] Sobczyk, Aleksandros. "Deterministic Complexity Analysis of Hermitian Eigenproblems." 52nd International Colloquium on Automata, Languages, and Programming (ICALP 2025). Schloss Dagstuhl–Leibniz-Zentrum für Informatik, 2025.
- [7] Ailon, Nir, and Bernard Chazelle. "The fast Johnson–Lindenstrauss transform and approximate nearest neighbors." SIAM Journal on computing 39.1 (2009): 302-322.
- [8] Tropp, Joel A. "Improved analysis of the subsampled randomized Hadamard transform." Advances in Adaptive Data Analysis 3.01n02 (2011): 115-126.
- [9] Tseng, Albert, et al. "QuIP $# $: Even Better LLM Quantization with Hadamard Incoherence and Lattice Codebooks." International Conference on Machine Learning. PMLR, 2024.
- [10] Ashkboos, Saleh, et al. "Quarot: Outlier-free 4-bit inference in rotated llms." Advances in Neural Information Processing Systems 37 (2024): 100213-100240.

**Questions:**

I have the following questions that would help me better understand the paper and to make a more informed final recomendation:
1) The goal of using the Hadamard transormation was to avoid outliers, so why do you need to apply also key-whitening?
2) It would be interesting to see the breakdown of how much memory is saved with the LoRA approximation and how much by having better quantization due to the removal of outliers.
3) How does the method compare computationally to prior work, in particular, to the additional references that I mentioned above? (exact SVD, Block-Krylov PCA, Lazy-SVD, other?). Would it benefit to replace the Power Iteration of HaPPI with a Block-Krylov iteration?
4) How does HaPPI compare to randomized Hadamard transform variants? Can you use randomness in HaPPI?
5) What do the bold-face numbers represent in Table 2? It is a bit confusing, for example, that in some cases the bold-face value is not the highest one in the corresponding column.
6) Line 2 of algorithm 1 is redundant. Line 3 of Algorithm 2 seems also redundant. Is it?
7) In Eq. (2), it is confusing that S is used both for the optimization variable, but also for the singular valuyes of $K^\top K$. Could you explain these steps in more detail, by using different matrices?

### Additional Feedback
I can mention the following additional points which could help the authors improve the manuscript. These points are not "necessary" to make my final recommendation.
- Some figures can be placed closer to where they are referenced. Especially Fig. 1, Fig. 2, and Fig. 3 show evaluation results, that belong to the analysis sections.
- Section 3.1 should be named "Singular **Value** Decomposition" (Singular Vector Decomposition is not wrong, but it is not the "established" acronym).
- In Fig. 3, it would be helpful to have a formula for MSE
- Between lines 185-187, I think you need to apply randomness to guarantee the outlier elimination (with high probability).
- A mathematical analysis (at least a basic one) of the algorithms, e.g., to justify their correctness, would greatly strengthen the presentation.
- Step 1 in Algorithm 3 seems redundant (is it?). I think you can directly call HaPPI(K,...), but I am not 100 percent sure.

**Details Of Ethics Concerns:**

No concerns.

---

> ### Author Response · Authors · 2025-11-21
>
> **[W1]** Mathematical analysis for HaPPI is required.
>
> We concur that rigorous mathematical proof would strengthen the paper, and this is marked as future work. HaPPI's primary contribution is its **practical effectiveness**: a stable and fast SVD approximation for outlier-heavy tensors.
>
> - **Hadamard Transform (HT):** HT improves numerical stability by uniformizing the tensor distribution (Figure 4b), preventing outlier amplification during $A^\top A$ computations in finite precision.
> - **PCA Initialization:** Accelerates convergence by initializing with principal component vectors from the covariance matrix, providing a 'near-optimal' starting point (Figure 2).
>
> **[W2]** Literature overview (Block-power iteration, Block-Krylov PCA, SVD=MM).
>
> Thank you for the valuable insight. We will enhance the Related Work section by including Block-power iteration, Block-Krylov PCA, and recent SVD=MM advancements to provide a stronger theoretical context.
>
> **[W3/Q4]** Randomized Hadamard (RHT) vs. Deterministic Hadamard (DHT).
>
> We acknowledge the potential benefits of Randomized Hadamard Transformation (RHT). However, minimizing latency is critical for real-time KV compression. RHT incurs additional overhead due to random sign generation and element-wise multiplication, whereas the deterministic Hadamard transform can be fully optimized using the Fast Walsh-Hadamard Transform (FWHT) with $O(N log N)$ complexity without any auxiliary costs, maximizing hardware acceleration efficiency.
>
> Our focus was on a deterministic Hadamard Transformation to maximize this hardware efficiency. We compared RHT and Deterministic Hadamard (DHT) in terms of MSE, FLOPs, and Latency. The results confirm that RHT not only yields a higher MSE but also increases FLOPs and Latency due to the overhead of generating and applying random signs.
>
> | Method | MSE | FLOPs | Latency |
> | --- | --- | --- | --- |
> | Random Hadamard | 1.8052 | 0.3051 | 0.0099 |
> | Deterministic Hadamard | 1.801 | 0.3009 | 0.0059 |
>
> **[W4]** Performance measurements: HaPPI-KV runtime breakdown.
>
> We agree that a breakdown of the end-to-end inference time is more insightful. We have performed the time decomposition for HaPPI-KV, separating the contributions of Key Whitening, HaPPI SVD, and Residual Quantization.
>
> | HaPPI-KV Components | Latency (ms) | Percentage (%) |
> | --- | --- | --- |
> | Key whitening | 3.7498 | 27.16 |
> | HaPPI SVD | 9.0791 | 65.76 |
> | Residual Quantization | 0.9775 | 7.08 |
> | Total | 13.8065 | 100 |
>
> **[W5/Q5]** Table/Figure descriptions.
>
> We will ensure all legends and axis titles are clearly defined. In our tables, bold-face numbers indicate the best performance among all compression methods, excluding the FP16 baseline. We will explicitly state this in the captions to avoid any confusion.
>
> **[W6]** Introduction: Better context/background.
>
> We will revise the Introduction to strengthen the background context. We will clearly articulate the importance of SVD approximation, the limitations of existing methods (Power Iteration, Randomized SVD) regarding outliers and speed, and how HaPPI's core ideas (HT + PCA Initialization) systematically address these challenges.
>
> **[Q1]** Why apply Key-whitening if Hadamard avoids outliers?
>
> HT and Key Whitening address outliers at **different levels**. HT mitigates internal **'value' outliers** for numerical stability. Key Whitening addresses the **cross-channel variance imbalance** (Figure 4a) of the K tensor itself. Both are crucial for maximizing compression efficiency.
>
> **[Q2]** Memory saving breakdown: Low-rank vs. Quantization.
>
> For HaPPI-KV (r=2, Q=2bit, 64k tokens), total memory is 15.49% of FP16. This usage is composed of: (1) The Low-rank matrix components (L×r + r×D), which account for ~10%, and (2) The 2-bit residual (L×D×(2/16)), which accounts for ~5.49%. The primary memory saving comes from the low-rank approximation, with the 2-bit residual quantization providing additional saving and crucial accuracy preservation.
>
> **[Q3]** Comparison to Block-Krylov & computational overhead.
>
> Thank you for this insightful suggestion. Block-Krylov Iteration is theoretically powerful for faster convergence. However, given the requirement for latency minimization in real-time KV cache compression, we chose the Power Iteration-based HaPPI for minimal overhead (one matrix multiplication per loop, deferred orthogonalization). We found this design to be latency-favorable in the current inference environment. We will explicitly discuss the potential of Block-Krylov methods as a promising direction.
>
> **[Q6]** Redundancy in Algorithm 1 & 2.
>
> - **Algorithm 1, Line 2:** We agree. Since $Q \leftarrow AP$ overwrites it on Line 7, this line is redundant and will be removed.
> - **Algorithm 2, Line 3:** This line is essential. It initializes $P$ with principal components from covariance ($C_H$), providing a **'near-optimal' starting point** for faster convergence (Figure 2).

---

> > ### Comment · Reviewer_hm4J · 2025-11-21
> > **Post-rebuttal assessment**
> >
> > I thank the authors for trying to address my concerns/questions as best as possible. I am quite familiar with the methods discussed here, and my current assessment is that the paper is not ready for publication, and I am not convinced that it will reach the required level by the time of publication. The paper seems some promising initial experimental results. At its current form, it can potentially be published as a workshop paper. However, if the authors want to pursue a full-paper in one of the top-tier AI conferences, the paper must undergo under a major revision. In that case, I strongly encourage the authors to carefully address all the topics that were raised by the reviewers, especially the major ones, such as a thorough discussion with the related literature **and** the related theory, and to re-submit to another upcoming conference (e.g., ICML). If they do not want to go through such a major revision, an alternative would be one of the ICLR workshops.

---

> > > ### Author Response · Authors · 2025-11-28
> > >
> > > We sincerely thank the reviewer for their continued engagement and detailed feedback. We respectfully believe that our work addresses the reviewer's concerns and meets the standards for publication at ICLR. Regarding the characterization of our results as "initial," we respectfully disagree. Our evaluation includes comprehensive results, encompassing Llama3-8B’s KV tensor to synthetic outlier tensor for HaPPI, three models (Llama3-8B, Mistral-7B, Qwen2-7B), nine benchmarks that cover CoT reasoning, long-context understanding, and synthetic tasks, and sequence lengths ranging from 4K to 64K tokens for HaPPI-KV. We also provide systematic ablation studies that isolate the contribution of each component. This level of experimental rigor is consistent with published works in this area.
> > >
> > >   On the discussion of related literature, we believe our paper provides a thorough review of relevant work, covering KV cache quantization methods (KIVI, KVQuant, WKVQuant), low-rank approaches (GEAR, PALU, SVDq), Hadamard-based techniques (QuaRot, SpinQuant, QuIP#, HALO), and SVD approximation algorithms (PowerSGD, randomized SVD). Furthermore, we would like to emphasize that HaPPI is fundamentally distinct from existing Hadamard-based methods. Prior works employ Hadamard transformations as a preprocessing step for quantization, explicitly aiming to spread outliers across channels. In contrast, HaPPI leverages the Hadamard transform for an entirely different purpose: improving the accuracy of low-rank approximation through PCA-based initialization in the frequency domain. The outlier mitigation in residuals is not our design goal but rather emerges naturally as a byproduct of operating in the frequency domain. This represents a novel perspective that has not been explored in the existing literature.
> > >
> > > We respectfully submit that our contributions meet the standards for a full paper at ICLR and welcome further discussion with the reviewer.

---

### Official Review · Reviewer_mWJC · 2025-10-31

**Soundness:** 4
**Presentation:** 4
**Contribution:** 4
**Rating:** 6
**Confidence:** 5

**Summary:**

The authors of this manuscript propose HaPPI, an approximate Truncated SVD algorithm that uses a Hadamard PCA-based Power Iteration to improve the accuracy of low-rank approximations while retaining efficiency. Building on this, they propose HaPPI-KV, a KV cache compression method that combines the HaPPI algorithm with key whitening and residual quantization. The authors claim this combined approach achieves state-of-the-art trade-offs in memory savings and model quality, outperforming prior methods like GEAR.

**Strengths:**

1- The paper is well-written and easy to follow.

2- The problem this work addresses, improving the efficiency and accuracy of SVD approximations, is very important in practice due to its large applications in different fields. Specially if the authors open source their code, it can be very impactful in the field.

3- The accuracy results reported for HaPPI-KV are very good, showing consistent and significant improvements over the GEAR baseline across multiple models and benchmarks.

4- The validation of the core HaPPI algorithm on LoRA initialization (Section 5.3) is a valuable addition, demonstrating that the improved approximation quality translates to better performance on downstream tasks.

**Weaknesses:**

1- The theoretical justifications for the improvements in HaPPI are limited. While the paper provides intuition (Hadamard for outliers, PCA for convergence), it lacks a formal analysis of approximation error bounds or convergence guarantees compared to standard power iteration or randomized SVD. Also, an analysis of the applicability of HaPPI on larger ranks (128, 256, 512) can be helpful.

2- The major contribution of this work is the HaPPI SVD approximation method, but its analysis feels narrow. The evaluation is largely focused on KV cache tensors. More analysis on matrices with different properties and distributions (e.g., various weight matrices from real models, synthetic data with controlled rank and outlier distributions) is essential to fully analyze the benefits and limitations of this method.

3- The timing results reported (Figure 6) are only for the compression function itself and only compare HaPPI-KV and GEAR. To understand the true practical overhead, a comparison against an uncompressed FP16 or BF16 benchmark (which would have zero compression latency) is needed to evaluate the impact on end-to-end inference time.

4- A detailed time decomposition of the inference process is missing. Showing the SVD time vs. the key whitening, quantization, and the rest of the model's computation can give more insight into the practical application and bottlenecks of HaPPI-KV.

5- The order of SVD and quantization in HaPPI-KV (SVD followed by residual quantization) is not well justified beyond a single ablation study result (Table 5). A deeper analysis of why this order is superior (e.g., analyzing the distribution of the residual) is needed, especially since the baseline (GEAR) quantizes first.

**Questions:**

1- The ablation study (Table 5) suggests applying HaPPI before quantization is better. How would quantizing the full tensor before applying the HaPPI low-rank approximation affect model quality? Could the authors provide a deeper analysis of why quantizing the residual is more effective?

2- How does the end-to-end inference latency of a model using HaPPI-KV perform in comparison to an uncompressed FP16 or BF16 benchmark, not just the latency of the compression function?

3- Could the authors provide a time decomposition of the inference latency when using HaPPI-KV? Specifically, what percentage of the total inference time is spent on the HaPPI SVD approximation versus the key whitening and residual quantization steps and other parts of the model?

4- How does the HaPPI algorithm's accuracy (MSE) and speed compare against other approximation methods (like `torch.svd_lowrank`) when applied to a wider variety of matrices, such as the main FFN weight matrices of a model, not just the KV cache?

---

> ### Author Response · Authors · 2025-11-21
>
> **[W1]** Limited theoretical justification & Applicability on larger ranks.
>
> We appreciate the demand for rigorous mathematical proof. The core contribution of HaPPI is presenting a **practical methodology** for stably and quickly approximating the SVD of outlier-heavy tensors. From the perspective of error bounds and convergence analysis, HaPPI fundamentally shares the same theoretical guarantees as standard Power Iteration, as it builds upon the same iterative framework. However, its key innovation lies in enhancing **numerical stability** and **convergence speed** via the Hadamard Transform and optimized initialization. Thus, while theoretical bounds remain identical, practical performance under finite precision is significantly improved.
>
> We also demonstrated that HaPPI outperforms torch.svd\_lowrank and standard Power Iteration by (1) uniformizing tensor distribution and (2) using PCA-based initialization for faster convergence (Figure 2).
>
> To confirm performance on larger ranks, we measured MSE on Llama3-8B's K state (Full rank = 128). Although the gap decreases at higher ranks, **HaPPI consistently achieves lower MSE**.
>
> | Rank | Loop | Real SVD | Power Iteration | HaPPI |
> | --- | --- | --- | --- | --- |
> | 16 | 2 | 0.8696 | 0.9703 | 0.8819 |
> | 32 | 2 | 0.4713 | 0.5438 | 0.4775 |
> | 64 | 2 | 0.1583 | 0.1929 | 0.1614 |
>
>
>
> **[W2, Q4]** Analysis on other matrices (weights, synthetic data) beyond KV cache.
>
> Thank you for suggesting a broader analysis. We measured MSE on various weight tensors from Llama3-8B's first block. The results show that the MSE reduction is less pronounced for weights compared to KV states. This confirms that weights lack large outliers, making traditional Power Iteration sufficient in such cases.
>
> | Tensor | Loop | Real SVD | Power Iteration | HaPPI |
> | --- | --- | --- | --- | --- |
> | Up-proj weight | 2 | 1.06E-04 | 1.08E-04 | 1.07E-04 |
> | Down-proj weight | 2 | 1.06E-04 | 1.08E-04 | 1.06E-04 |
> | Attention W_Q | 2 | 1.92E-04 | 2.01E-04 | 1.92E-04 |
>
> We also tested synthetic tensors with varying outlier ratios (Student's T-distribution). The performance gain from HaPPI diminishes as the Degree of Freedom (DF) increases (fewer outliers). This clearly demonstrates that **HaPPI is most effective in scenarios with severe outliers**.
>
> | Degree of Freedom | Loop | Real SVD | Power Iteration | HaPPI |
> | --- | --- | --- | --- | --- |
> | 3 | 2 | 2.3631 | 2.4857 | 2.3930 |
> | 10 | 2 | 1.0271 | 1.0698 | 1.0411 |
> | 30 | 2 | 0.8806 | 0.9075 | 0.8925 |
>
> ---
>
> **[W3, Q2, W4, Q3]** Practical overhead comparison (vs FP16) and latency decomposition.
>
> We agree on the need for a comprehensive comparison. We measured **end-to-end inference latency** of Llama3-8B on GSM8K (seq len $\approx$ 4096, rank=4) on an A100 GPU, comparing HaPPI-KV against FP16 and GEAR.
>
> | Method | Generation (s) | Compression (s) | Total (s) |
> | --- | --- | --- | --- |
> | FP16  | 7.3138 | - | 7.3138 |
> | GEAR |  7.3138 | 1.7779 | 9.0917 |
> | HaPPI-KV |  7.3138 | 1.8172 | 9.1310 |
>
> Additionally, we measured the decomposition of the HaPPI-KV compression latency of one attention layer:
>
> | HaPPI-KV Components | Latency (ms) | Percentage (%) |
> | --- | --- | --- |
> | Key whitening | 3.7498 | 27.16 |
> | HaPPI SVD | 9.0791 | 65.76 |
> | Residual Quantization | 0.9775 | 7.08 |
> | Total | 13.8065 | 100 |
>
> HaPPI SVD accounts for 65.76%, indicating potential for optimization via kernel fusion. Additionally, memory reduction allows for larger batch sizes, potentially enhancing system throughput in high-load scenarios.
>
>
>
> **[Q1]** Why is applying HaPPI before quantization (residual quantization) more effective?
>
> We compared MSE for Quant-SVD (GEAR) vs. SVD-Quant (HaPPI-KV). Results show HaPPI-KV achieves lower overall Reconstruction MSE.
>
> | Method | Compressed tensor MSE | Residual MSE | Reconstruction MSE |
> | --- | --- | --- | --- |
> | Quant-SVD (GEAR) | 0.7124 | 0.6883 | 0.6883 |
> | SVD-Quant (HaPPI-KV) | 2.8291 | 0.6808 | 0.6808 |
>
> The reconstruction MSE of the entire pipeline is equal to the Residual MSE, as shown in the derivation below:
>
> 1. **Compressed MSE** = $MSE(X, C) = E[(X - C)^2] = E[R^2]$
> 2. **Residual MSE** = $MSE(R, R') = E[(R - R')^2]$
> 3. **Reconstruction MSE** = $MSE(X, X') = MSE(X, C + R') = E[(X - C - R')^2] = E[(R - R')^2]$ = **Residual MSE**
>
> The reconstruction MSE equals the Residual MSE (derived below). In HaPPI-KV, the initial SVD removes high-magnitude outliers, significantly reducing the residual's dynamic range. This makes subsequent quantization more effective, improving overall reconstruction quality.

---

> ### Comment · Reviewer_mWJC · 2025-11-26
>
> I thank the reviewers for providing the additional details. Specially, the time breakdown results are very insightful for detecting and improving the bottlenecks. The theoretical justification for the performance of HaaPPI is still missing, and such justifications can help with understanding and improving proposed method further.
>
> I'll keep my positive evaluation of this work, though other important concerns have been raised by other reviewers.

---

### Official Review · Reviewer_QerW · 2025-11-01

**Soundness:** 3
**Presentation:** 2
**Contribution:** 3
**Rating:** 6
**Confidence:** 3

**Summary:**

This paper proposes a new SVD algorithm that improves upon the conventional power iteration method by making the following changes:
- Applies a Hadamard Transform prior to power iteration.
- Uses the covariance matrix of the transformed tensor for initializing the power iteration.

The paper then applies this SVD method to KV cache compression, with additional improvements specific to KV cache such as key whitening and residual quantization.

**Strengths:**

- A novel SVD algorithm well tailored for capturing the low-rank structure of the KV cache.
- Strong empirical validation compared to a sufficiently wide range of other KV cache compression methods.

**Weaknesses:**

**[W1]** In Table 2, showing the compression ratios of different methods and configurations would strengthen the claim. This can also be presented as an accuracy vs. memory usage plot.

**[W2]** The plotted figures are too small and therefore hard to read, especially the font sizes for the legends and axis titles.

Minor comments that did not affect the score:
- L469: Double-check the number for GEAR’s latency measurement.

**Questions:**

**[Q1]** Would the strong performance hold for smaller and larger model scales?

**[Q2]** How would the method perform for reasoning models such as the Qwen3 model family on even longer generation tasks, up to the scale of ~10k tokens? Demonstrating this would highlight the method’s robustness under long-generation scenarios, which are not captured by current CoT tasks or long-context retrieval tasks.

---

> ### Author Response · Authors · 2025-11-21
>
> **[W1]** In Table 2, showing the compression ratios of different methods and configurations would strengthen the claim. This can also be presented as an accuracy vs. memory usage plot.
>
> Thank you for the suggestion. While the relative memory usage for each method is already noted in the legend of Figure 1 (e.g., HaPPI-KV(r=2) 15.73%, GEAR(r=4) 21.53%), we agree that including this information in Table 2 will significantly enhance clarity. We will revise Table 2 to present both accuracy and memory efficiency side-by-side, allowing readers to compare these two critical metrics at a glance.
>
>
> **[W2]** The plotted figures are too small and therefore hard to read, especially the font sizes for the legends and axis titles.
>
> We apologize for the low readability of the current figures. For the camera-ready version, we will significantly improve the quality of all figures (including Figure 1, 2, 3, and 6) by increasing the resolution and enlarging the font sizes for the legends and axis titles.
>
>
> **[W3]** L469: Double-check the number for GEAR’s latency measurement.
>
> Thank you for catching this error. We have double-checked the latency measurement for GEAR in the 65,536 token scenario and confirmed the error (81.43ms should be **142.8ms**). We have corrected this number in the revised PDF version of the paper.
>
>
> **[Q1]** Would the strong performance hold for smaller and larger model scales?
>
> We expect HaPPI to demonstrate consistent performance improvements across various model scales (both smaller and larger than Llama3-8B). HaPPI is a general SVD approximation algorithm that leverages the low-rank structure of the tensor and primarily depends on the intrinsic properties of the KV cache tensor itself (e.g., outlier distribution, intrinsic rank), rather than the absolute size of the model.
>
> We conducted a scaling experiment to compare compression performance across different model sizes. The results below confirm our expectation.
>
> | Model | Method | GSM8K ACC |
> | --- | --- | --- |
> | Llama3-1B | FP16 | 44.43 |
> | Llama3-1B | GEAR | 32.75 |
> | Llama3-1B | HaPPI-KV | 40.94 |
> | Llama2-13B | FP16 | 32.6 |
> | Llama2-13B | GEAR | 31.31 |
> | Llama2-13B | HaPPI-KV | 31.41 |
>
>
>
> **[Q2]** How would the method perform for reasoning models such as the Qwen3 model family on even longer generation tasks, up to the scale of ~10k tokens?
>
> Due to the time constraints of the first rebuttal phase (ending November 20th), we could not complete the experiments on the extended 10k token sequence in time. However, we fully recognize the importance of evaluating long-context reasoning capabilities. We are currently running these experiments and will update the results as soon as they are completed.

---

> > ### Comment · Reviewer_QerW · 2025-11-28
> >
> > Thank you for the rebuttal and the additional experimental results. I appreciate the planned improvements to the presentation, and I encourage the authors to incorporate these changes as well as the long-context evaluation results into the camera-ready version if the paper is accepted. I will keep my overall score unchanged.

---

### Meta-Review · Area_Chair_XkLB · 2025-12-23

**Summary:**

The paper introduces HaPPI, an approximate SVD algorithm utilizing Hadamard transforms and PCA-based initialization, and HaPPI-KV, a pipeline for compressing LLM KV caches. Empirical evaluations on Llama3, Mistral, and Qwen2 suggest the method outperforms existing compression techniques like GEAR in retaining model accuracy across reasoning and long-context tasks.

Strengths
- Reviewers acknowledged that the method demonstrates strong empirical results.
- The inclusion of LoRA initialization experiments and scaling laws (in the rebuttal) provided evidence of the method's applicability.

Weaknesses
- Multiple reviewers (z8pu, 27GN, hm4J) raised concerns regarding the theoretical justification of the algorithm.
- While the authors argue in the rebuttal that the benefits stem from numerical stability in finite precision (mitigating outlier amplification), the paper provides no rigorous error analysis supporting this claim.
- The algorithmic core is identified by reviewers as a variation of existing Block Power Iteration or RSVD methods so novelty is limited

Overall, the paper is interesting but it is not ready for publication in the current form.

**Reviewer Concerns:**

Reviewer mWJC raised concerns about theoretical contribution of the work that have not been addressed.

As mentioned by reviewer hm4J, the concerns raised by the reviewer have been only partially addressed

**Reviewer Scores:**

Reviewer QerW would probably not change the score.

Reviewer mWJC would probably not change the score.

Reviewer hm4J would probably not change the score.

Maybe reviewer 27GN would have slightly increased the score.

Reviewer z8pu  would probably not change the score.

---

### Decision · Program_Chairs · 2026-01-26

Reject